# Short-Term Drift Prediction of Multi-Functional Buoys in Inland Rivers Based on Deep Learning

**DOI:** 10.3390/s22145120

**Published:** 2022-07-07

**Authors:** Fei Zeng, Hongri Ou, Qing Wu

**Affiliations:** 1Key Laboratory of Metallurgical Equipment and Control Technology of Ministry of Education, Wuhan University of Science and Technology, Wuhan 430081, China; hongriou@wust.edu.cn; 2Key Laboratory for Port Handling Technology Ministry of Communication, Wuhan University of Technology, Wuhan 430063, China; wq@whut.edu.cn

**Keywords:** inland waterway, multi-functional buoy, drift prediction, deep learning, hybrid model, time series prediction, ship navigation

## Abstract

The multi-functional buoy is an important facility for assisting the navigation of inland waterway ships. Therefore, real-time tracking of its position is an essential process to ensure the safety of ship navigation. Aiming at the problem of the low accuracy of multi-functional buoy drift prediction, an integrated deep learning model incorporating the attention mechanism and ResNet-GRU (RGA) to predict short-term drift values of buoys is proposed. The model has the strong feature expression capability of ResNet and the temporal memory capability of GRU, and the attention mechanism can capture important information adaptively, which can solve the nonlinear time series drift prediction problem well. In this paper, the data collected from multi-functional buoy #4 at Nantong anchorage No. 2 in the Yangtze River waters in China were studied as an example, and first linear interpolation was used for filling in missing values; then, input variables were selected based on Pearson correlation analysis, and finally, the model structure was designed for training and testing. The experimental results show that the mean square error, mean absolute error, root mean square error and mean percentage error of the RGA model on the test set are 5.113036, 1.609969, 2.261202 and 15.575886, respectively, which are significantly better than other models. This study provides a new idea for predicting the short-term drift of multi-functional buoys, which is helpful for their tracking and management.

## 1. Introduction

Multi-functional buoys are critical equipment for indicating waterway or anchorage boundaries and obstructions in inland rivers, and they are of great significance to the safe navigation of ships [1]. However, due to the influence of external or human factors, such as severe weather, collision or towing of passing ships, failure of mooring devices, etc., multi-functional buoys may cause the phenomenon of sinker displacement and anchor chain fracture, resulting in abnormal deviation of its position. Once the drift value is too large, wrong navigation aid information will be issued, which will result in great potential safety hazards to the navigation of passing ships. Therefore, it is important to predict the drift position of the buoys accurately in real time to ensure the safety of ship navigation.

At present, the prediction of the buoy drift position is mainly achieved by constructing explicit mathematical models [2,3,4,5] or based on data-driven methods. Among them, the method of constructing explicit mathematical models is simple, but the prediction results cannot satisfy accuracy requirements, because it cannot simulate the change mechanism of buoy drift with the influence of wind, currents, waves and other physical factors [6]. However, data-driven methods construct approximate implicit models to approach the real situation based on a large amount of historical data, possessing higher accuracy, and can be divided into statistical and machine learning methods [7]. The statistical learning method is based on sensor monitoring data and is used to analyze and predict with probabilistic statistical algorithms. Chen [8] used an autoregressive integrated moving average model (ARIMA) to predict the displacement distance of buoys in the Yangtze Estuary deep-water waterway for the next 24 h and realized the displacement early warning. Wu et al. [9] proposed a multiplicative seasonal ARIMA model to predict the future shift distance of buoy No. 1 in Meizhou Bay. Han et al. [10] constructed an adaptive Kalman filter to estimate the drift trajectory of the buoy. Although the above probabilistic statistical model can achieve a certain prediction effect, it requires the high stability of buoy data and has limited prediction ability for nonlinear data. Machine learning-based methods are more robust for data prediction, and some typical models are insensitive to data with nonlinear characteristics, such as support vector machine (SVM), decision trees and artificial neural networks (ANNs). However, SVM is suitable for dealing with small-batch data, and the decision tree is prone to overfitting, while ANN has the self-learning capability to solve nonlinear prediction problems for large-scale data [11], and thus, it has attracted enormous interest from scholars in buoy drift prediction. Li et al. [12] used a BP neural network (BPNN) to predict the drift position of buoy T10 in the Tonggu waterway. Fang [13] established a fractional-order gradient descent with the momentum RBF neural network algorithm to predict the drift value of buoy #4 in Xiamen port. Xu [14] proposed an improved complex-valued neural network to implement the drift position prediction of buoys 11A and No. 2 in Meizhou Bay. These ANN-based models provide better prediction results than most statistical learning models, but the input features only consider one or two variables among the wind speed, latitude and longitude, and tide values. In addition, it is difficult to accurately determine the change in the amount of buoy drift because the multi-functional buoy’s drift has typical time series characteristics, and the above study did not consider feature extraction and contextual information correlation of the time series data.

Nevertheless, prediction models based on deep learning methods exhibit better data fitting and prediction performance because they can automatically extract features and correlate contextual information [15]. Typical deep learning models include convolutional neural networks (CNNs) and recurrent neural networks (RNNs) [16]. CNNs are used to mine the local features of data through the sparsity of connections and parameter sharing, which significantly reduces the network parameters and makes the model easier to train [17]. For this reason, CNN-based models [18,19,20] have achieved excellent performance in image processing. Buoy drift values and other hydrometeorological data are converted to a grid along the time axis and can be considered a single-channel image, so CNN can be used to extract its local features. In addition, RNNs are excellent at associating contextual information on the time axis [21], but they create the vanishing or exploding gradient problem when the sequence information is long. To solve this problem, Hochreiter et al. [22] proposed a long short-term memory network (LSTM) by improving the structure of RNN, which introduces a “gate mechanism” to update the state. However, it has the disadvantage of more parameters, which increases the training cost [23]. Therefore, Chung et al. [24] further improved the LSTM model and then proposed the gated recurrent unit (GRU), which reduced the three gates in the LSTM to two, reducing the computational effort and providing little difference in accuracy [25]. Because of the significant time series characteristics of buoy telemetry data, the GRU can be used to correlate contextual information. However, as the length of the buoy sequence data increases, important information may be lost, as the GRU is not excellent at capturing long-term dependencies [26]. In recent years, the attention mechanism has been used to capture long-term dependencies, because it can assign greater weights to important information and make each output retain all of the input information [27]. Therefore, the attention mechanism can be used to assign weights to the input information adaptively so as to effectively improve the accuracy of buoy drift prediction.

In recent time series prediction studies, such as traffic flow prediction [28], soil water content prediction [29] and load prediction [30], experimental results have shown that hybrid model-based prediction methods combine the advantages of two or more models and provide better prediction performance than a single model. Nevertheless, for buoy drift prediction, Pan et al. [31] used a GRU-Attention model to achieve buoy drift value prediction in the lower Yangtze River section. However, the input features only involved water level values, latitude and longitude, did not consider the effects of wind speed, water velocity and tidal values, and did not consider the inclusion of CNN to capture short-term dependence, so the prediction accuracy could be improved.

In addition, the long-term prediction of multi-functional buoy drift is influenced by external factors and human factors, so it is difficult to achieve ideal results. To this end, a hybrid model for the short-term drift prediction of inland waterway buoys incorporating ResNet-GRU and the attention mechanism is proposed in this paper, which combines the local feature extraction ability of ResNet, the time series memory learning ability of the GRU and the advantage of weight assignment of the attention mechanism, and the latitude and longitude, water velocity, tidal value, wind speed and other factors are used as the inputs of the model features for training so as to accurately predict the drift of multi-functional buoys. The main objectives of this study were:(1)To construct the structure of the RGA hybrid model and determine optimal parameter settings and then improve the prediction accuracy of short-term drift of inland waterway buoys;(2)To analyze the effects of different hydrometeorological parameters on the variation in buoy drift and select input features for the prediction model;(3)To evaluate the drift prediction performance of the RGA model compared with other baseline models.

The layout of the rest of the paper is as follows: Section 2 presents the RGA model architecture and introduces the methods of ResNet, GRU and attention layers. Section 3 describes the source of the data and the processing of the dataset, followed by the variable correlation analysis. Section 4 shows a comparison of all models and a detailed analysis of the results. Section 5 provides the conclusions and limitations of this paper and proposes future research work.

## 2. An RGA Model for Multi-Functional Buoy Drift Prediction

Multi-functional buoy drift has inertia and continuity and is in line with the characteristics of time series data, so an RGA model for multi-functional buoy drift prediction was designed, which has strong capabilities of feature extraction and time series mining.

The RGA model architecture is shown in Figure 1. The model contains an input layer (Input), a ResNet layer, a fully connected layer, two GRU layers, an attention layer, a flatten layer and an output layer.

First, the data are stitched into a grid along the time axis, as shown in Figure 1a, where *N* is the number of features, *T* is the length of the time step, which is in units of h, *t* denotes the current moment. Since the prediction method in this paper is to use the previous *T* steps (*t* − (*T* − 1), *t* − (*T* − 2), …, *t*) to predict the drift value at the next moment (*t* + 1), so the grid is converted into a matrix and fed into the model, and the local features are extracted through the ResNet layer. Then, the output of the ResNet layer is sent to the fully connected layer, which aims to adjust the size of the output vector and enhance the nonlinear mapping capability of the model. Next, superposing of two GRU layers is used to capture the change trend of the sequence and learn higher-level time-domain feature representations, and the output of its fully connected layer is fed to the first GRU layer. The hidden states of the second GRU layer are then used as input to the attention layer, which is used to learn the importance of each hidden state. Finally, the output of the attention layer is flattened and then connected to the output layer.

### 2.1. ResNet Layer

Considering that the multi-functional buoy dataset is converted to a grid along the time axis and can be considered a single-channel image, CNN is used as a network component to improve the drift prediction accuracy. Previous studies have shown that deepening the number of network layers can extract richer features [32]. However, deeper networks are not always ideal, owing to the tendency of backpropagation to produce vanishing or exploding gradients [33]. Therefore, in 2015, Kaiming He et al. [34] proposed ResNet containing “skip connections”, which is composed of multiple stacked residual blocks, as shown in Figure 2. The computational procedure is:*h*(*x*) = *F*(*x* + *w_i_*)(1)
where *x* and *h*(*x*) are the input and output, and *F*(*x* + *w_i_*) denotes the mapping function to be learned by the network.

In this paper, a modified residual block is used to extract the local features of the input data [35], as shown in Figure 3. In the improved version, the gradient can be quickly transferred to the previous layer through the “skip connection”, which solves the vanishing or exploding gradient problem of the deep network and has better performance. Considering the complexity of the buoy drift data, a two-dimensional (2D) convolutional layer was used as the structure of the residual block to guarantee the stability of the prediction effect, and the strong feature expression capability of the 2D convolutional layer is conducive to automatically extract common features at different levels of the time series data. Figure 4 shows an example of a residual block with 64 convolutional kernels used, where “CONV” denotes the 2D convolutional layer, “BN” represents the batch normalization layer, and “Relu” denotes the Relu activation layer.

### 2.2. GRU Layer

To further mine the time series features of the input data, the GRU layer is used to receive the output of the fully connected layer after ResNet. Since the GRU layer requires 3D input, the 2D output of the fully connected layer is converted into 3D input by a Reshape operation before receiving the input information. The RGA model uses two stacked connected GRU layers to learn more advanced time series features and then go on to accurately predict the trend of the buoy drift. In addition, in order to prevent the model from overfitting, a spatial decay operation is used in the second GRU layer to discard 20% of the feature maps. Figure 5 is a schematic diagram of the structure of the GRU unit.

In the GRU, update gate *z_t_* and reset gate *r_t_* are the core modules. The update gate determines the extent to which the state information from the previous moment is retained in the current state, the reset gate controls the amount of information written to the candidate hidden state at the previous moment, and the output *h_t_* of the hidden layer at time *t* is calculated as follows [36]:(2)$$r_{t}=\sigma (x_{t}W_{xr}+h_{t-1}W_{hr}+b_{r})$$
(3)$$z_{t}=\sigma (x_{t}W_{xz}+h_{t-1}W_{hz}+b_{z})$$
(4)$${\tilde{h}}_{t}=\tanh(x_{t}W_{xh}+(r_{t}\times h_{t-1})W_{hh}+b_{h})$$
(5)$$h_{t}=z_{t}\times h_{t-1}+(I-z_{t})\times {\tilde{h}}_{t}$$
where, for a given time *t*, *x_t_*, *h_t_*_−1_ and ${\tilde{h}}_{t}$ are the input vector, the hidden layer state at time *t* − 1 and the candidate hidden state, respectively; *W_xr_*, *W_hr_*, *W_xz_*, *W_hz_*, *W_xh_* and *W_hh_* are the weight parameters between *r_t_*, *z_t_*, ${\tilde{h}}_{t}$ and *x_t_*, *h_t_*_−1_, respectively; *b_r_*, *b_z_* and *b_h_* are *r*, *z_t_* and ${\tilde{h}}_{t}$ bias terms, respectively; I denotes the unit matrix; × is the Hadamard product (product by elements) operator; σ is a sigmoid activation function; tanh denotes the tanh activation function; and the mathematical equation is described as follows:(6)$$\sigma (x)=\frac{1}{1+e^{-x}}$$
(7)$$tanh\left(x\right)=\frac{e^{x}-e^{-x}}{e^{x}+e^{-x}}$$

### 2.3. Attention Layer

To focus on the influence of different input features on the drift prediction of the multi-functional buoy, the attention mechanism is introduced into the RGA model to improve the drift prediction accuracy. The attention layer first receives the output of the second GRU layer and then automatically learns the importance of each hidden state, focusing on important information and suppressing useless information; its structure is shown in Figure 6.

The attention layer can be simply described as weighted summation, which first calculates the matching degree between each hidden state of the input by using a scoring function S and inputs the matching degree score into the softmax function to obtain the attention weights. Then, the weights and the input features are multiplied correspondingly, and finally, they are summed to obtain the output. The calculation process can be described as [37]:(8)$$e_{t}=\tanh(wh_{t}+b)$$
(9)$$\alpha_{t}=\frac{\exp(e_{t})}{\sum_{j=1}^{T}\exp(e_{j})}$$
(10)$$z_{t}=\sum_{t=1}^{T}\alpha_{t}h_{t}$$
where *h_t_* is the input of the attention layer at time *t*, that is, the output of the GRU, *w* is the weight matrix, *b* is the bias, *e_t_* is the value of the attention probability distribution at time *t*, *α_t_* is the final weight assigned to each value of the input, and *z_t_* is the output of the attention layer at time *t*.

## 3. Data Acquisition and Preprocessing

### 3.1. Data Acquisition

Multi-functional buoy #4 is near the mouth of the Donghai Sea in Shanghai, located in the intersection area of the Tonglu Canal and the Yangtze River, where there is a large ship traffic volume and a complex navigation environment. Especially at night or in bad weather, a variety of complicated ships will reach anchorage No. 2 in Nantong for anchoring, which increases the risk of mutual collision. Therefore, in this study, multi-functional buoy #4 at Nantong anchorage No. 2 near the main waterway of the Yangtze River was selected as the object, and its layout base point is 120°48′18.7272″ E, 31°59′56.5368″ N. Then, the telemetry data of the multi-functional buoy and the tidal observation data of Tiansheng Harbor within the period from September 5th, 2019, to October 8th, 2021, were collected. The data collection interval was 1 h, thus forming a time series dataset. Among them, the multi-functional buoy telemetry data are drift value, longitude, latitude, water velocity, water direction, wind speed, wind direction and wind scale, which were acquired by a positioning chip, a high-precision Geolu RSS-2-300W non-contact radar current-meter and an FWS 200 two-element wind speed and direction sensor, respectively. The data collected by these sensors were transmitted by the data bus to the telemetering and telecontrol terminal for packaging and transmitted to the DTU (Data Transfer Unit) through a 4G network, which converts the network data into serial data and sends them to the hydrometeorological system server, which unpacks and then stores the data packets and serves as a data service platform to provide access interfaces for other maritime intranet and public network applications. In this study, the interface was accessed through the JavaScript front-end development language, and then data acquisition software was created, which can be used to export multi-functional buoy telemetry data. However, the tidal observation data of Tiansheng Harbor are tidal values, which were obtained by the global tidal prediction service platform (http://global-tide.nmdis.org.cn/) (accessed on 7 December 2021) query. Table 1 shows the abbreviations and units of each variable collected.

### 3.2. Data Preprocessing and Analysis

#### 3.2.1. Data Preprocessing

Missing values are inevitable in the dataset owing to uncontrollable factors, such as sensor failure or inspection and maintenance by managers. Considering that the time series data change slowly in a short period of time, linear interpolation was used for filling in missing values. Assuming that the missing sensor value at moment *t_n_* is *x_n_*, it can be obtained by linear interpolation of the measured values *x_n_*_−1_ and *x_n_*_+1_ at the moments before and after the missing value [38]:(11)$$x_{n}=\frac{t_{n+1}-t_{n}}{t_{n+1}-t_{n-1}}x_{n-1}+\frac{t_{n}-t_{n-1}}{t_{n+1}-t_{n-1}}x_{n+1}$$
where *t_n_*_−1_ means 1 h before *t_n_*, and *x_n_*_+1_ means 1 h after *t_n_*.

To facilitate training, some features in the data need to be quantified, such as the water direction (incoming water and outgoing water), wind direction f (east, north, west, south, etc.) and wind scale (no wind, soft wind, light wind, breeze, strong wind, etc.).

In addition, owing to the varying ranges of different data values, in order to enhance the prediction accuracy and convergence speed of the model, in this study, the max–min normalization method was used to constrain each feature data value within [0, 1]. The equation is as follows:(12)$$x\prime =\frac{x-x_{min}}{x_{max}-x_{min}}$$
where *x’* is the normalized data, *x* is the original data, and *x_min_* and *x_max_* are the minimum and maximum values in the sample data, respectively.

#### 3.2.2. Correlation Analysis of Eigenvalue

After data cleaning, the Pearson correlation coefficient R was used to assess the strength of the correlations among the eight hydrometeorological varaibles (Lng, Lat, Wv, Wad, Wsp, Wid, Wsc and Ht) and the multi-functional buoy drift value (Drd), and the results are shown in Figure 7, where the larger the value of |R|, the higher the degree of correlation.

As can be seen in Figure 7, the correlation between wind scale (Wsc) and Drd is the weakest; thus, considering that Wsc is derived from wind speed division, the effect of Wsc factor is ignored for drift prediction. 

However, the correlation between the tidal value (Ht) and Drd is the strongest, because the buoy is tethered in the water by the sinker and anchor chain and is deployed near Tiansheng Harbor of the Yangtze River; the tide of Tiansheng Harbor is an irregular semidiurnal tide, the tide level shows a trend of two rises and two falls daily in each flood and ebb-tide period, and the buoy is influenced by the tide and moves reciprocally along the waterway direction, so there is a significant correlation between the two.

In addition, compared with the Ht variables, the correlations of wind speed (Wsp) and water velocity (Wv) are not significant, but they are important factors affecting drift. If Wsp is dominant, the drift direction of the buoy is mainly inclined to the downwind direction, and when the Wv dominates, it will be offset along the waterway axis.

## 4. Experiment

The hardware and software platform configurations used in the experiments are shown in Table 2.

### 4.1. The Metrics of Model Performance

In this paper, mean square error (MSE), mean absolute error (MAE), root mean square error (RMSE) and mean absolute percentage error (MAPE) are used to evaluate the performance of the model, and the smaller the value, the higher the accuracy of the model prediction. The four metrics can be formulated as follows [39]:(13)$$\mathrm{MSE}=\frac{1}{n}\sum_{i=1}^{n}{\left(y_{i}-{\hat{y}}_{i}\right)}^{2}$$
(14)$$\mathrm{MAE}=\frac{1}{n}\sum_{i=1}^{n}\left|y_{i}-{\hat{y}}_{i}\right|$$
(15)$$\mathrm{RMSE}=\sqrt{\frac{1}{n}{\sum_{i=1}^{n}\left(y_{i}-{\hat{y}}_{i}\right)}^{2}}$$
(16)$$\mathrm{MAPE}=\frac{1}{n}\sum_{i=1}^{n}\frac{\left|y_{i}-{\hat{y}}_{i}\right|}{{\hat{y}}_{i}}$$
where *n* is the number of samples, *y_i_* is the true value, and ${\hat{y}}_{i}$ is the predicted value.

### 4.2. Network Model Training

The input dimensionality of the model is 8 dimensions, including 8 hydrometeorological variables (Lng, Lat, Wv, Wad, Wsp, Wid, Ht and Drd), and the output is Drd. About the first 80% of the data were used as the training set, and the last 20% were used as the test set. For the training process, the original training dataset was divided into an 80% training set and a 20% validation set to calibrate the model.

To determine the optimal model hyperparameters, the model was trained several times. During the process of training, the early stopping method was adopted to preserve the optimal weights and avoid overfitting so as to obtain better generalization performance. Among them, MSE was used as the loss function, Adam was utilized as the optimizer, the learning rate was 0.0001, and the batch size and epoch were set to 256 and 250, respectively. In addition, the larger the time step, the longer the training time and the greater the noise; hence, the time step was set to 12.

### 4.3. Parameter Settings of RGA Model

The RGA model mainly consists of the ResNet layer, GRU layer and attention layer. The parameter settings of the ResNet layer and GRU layer have a significant impact on the prediction performance of the RGA model, in which the number of residual block stacks, the number of feature maps and the size of convolutional kernels need to be determined for the ResNet layer, and the GRU needs to determine the number of neurons in each layer. Because the number of datasets in this paper is small, in order to extract high-level abstract features faster with fewer layers and avoid losing important features, the number of residual block stacks was set to 2. To determine the optimal model parameters, several experiments were conducted on a selection of model parameters, such as the number of feature maps, the size of the convolutional kernel, the number of GRU neurons, etc., and then the model weights with the optimal experimental results were saved for prediction. The optimal RGA model parameters are set as follows:

The numbers of neurons in the fully connected layer and the output layer are 64 and 1, respectively. The main parameters of ResNet and GRU are set as shown in Table 3 and Table 4, where the first convolutional layer kernel size of the 2nd residual block is 1 × 1, and the stride is 2. In addition, a timer is set in the code to record the time, which provides a reference for evaluating the performance of the model.

### 4.4. Parameter Settings of Other Comparison Models

To verify the effectiveness of the RGA model proposed in this paper for the short-term drift prediction of multi-functional buoys, the following models were used as benchmarks for comparison experiments, and the parameters were set as follows.

**BPNN:** The BPNN has two hidden layers; the numbers of neurons are 24 and 48, respectively, epoch = 300, and other required parameters are consistent with the RGA model proposed in this paper so as to ensure comparative rationality.

**SVR:** The common kernel functions of the SVR model include the linear kernel function, radial basis function (RBF), etc., among which the RBF kernel function is widely used with fewer parameters and higher efficiency. Therefore, the RBF kernel function was chosen for the SVR model, and the values of the regularization parameter (C) and kernel coefficient (gamma) were determined by a grid search, with C chosen from {1, 10, 100, 1000, 10,000} and gamma chosen from {0.01, 0.1, 1, 10, 100}. According to the model training results, the optimal C and gamma are set to 10 and 0.01, respectively.

**ResNet:** The model has the same parameter settings as the RGA model, except that the GRU layer and attention layer are not introduced, but it has 2 fully connected layers after the flatten layer with 1024 and 512 neurons and an output layer after the fully connected layer with 1 neuron.

**CNN:** Its structure is the same as that of the ResNet model, except that it does not introduce the “skip connection”.

**GRU:** The parameter settings are the same as those of the RGA model, except that there are no ResNet and attention layers.

**LSTM:** The model has two stacked connected LSTM layers, and the parameter settings are consistent with the GRU model.

**ResNet-GRU:** The parameter settings are the same as those of the RGA model, except that the attention layer is not introduced.

**GRU-Attention:** Similarly, the parameter settings are the same as those of the RGA model, except that the ResNet layer is not introduced.

### 4.5. Experimental Results and Discussion

After determining the hyperparameters, the proposed RGA model was trained, and then its learning curve of training and validation loss versus the number of epochs was plotted, as shown in Figure 8. The results show that the training loss and validation loss of the RGA model decrease with the increase in the number of epochs; the decrease in training loss indicates that its fit is improving, and the decrease in validation loss indicates that the model is not overfitted. In addition, the two loss curves are very close to each other, indicating that the model has a better generalization performance.

The RGA model was compared with the eight models mentioned in Section 4.4, and the results are shown in Table 5 and Figure 9, from which it can be seen that most of the deep learning (DL) models outperform the traditional machine learning (ML) models, while the proposed RGA model has the best performance. The model reduces the MSE, MAE, RMSE and MAPE metrics by 21.1%, 12.5%, 11.2% and 0.03%, respectively, compared to the ResNet-GRU model and by 10.1%, 7.1%, 5.2% and 7.9%, respectively, compared to the GRU-Attention model. Similarly, the performance of the GRU-Attention model is better than that of the GRU model, which fully shows the success of introducing the attention mechanism into the model, as it assigns different weights to each hidden state by calculating the attention score and effectively extracts useful information.

It is worth noting that in our dataset, the ResNet-GRU model (MAE = 1.841023; RMSE = 2.545639) has lower predictive power than the GRU model (MAE = 1.735634; RMSE = 2.426921), probably because ResNet-GRU is less robust in the face of weaker nonlinear data [40]. Compared with CNN and ResNet, the performance of LSTM and the GRU is better, indicating the existence of complex time series characteristics of our data, and the advantage of LSTM and GRU is that they are excellent at capturing it; the prediction performance of GRU is slightly lower than that of LSTM, but the training time of GRU is 225.7797 s lower than the 233.0197 s of LSTM, which is because the GRU has fewer parameters, so the training time is shorter and the learning ability is slightly weaker than those of LSTM, but the overall difference in error between the two is not significant. In addition, the MSE, MAE, RMSE and MAPE metrics of the ResNet model were reduced by 40.6%, 27.3%, 22.9% and 10.1% compared to the CNN without the “skip connection”. This significant improvement in prediction performance demonstrates the advantage of ResNet for drift prediction.

Furthermore, the prediction performance of SVR and BPNN is worse than that of the models with time series feature learning ability. Although they can solve the problem of nonlinear regression, they cannot capture the correlation in time series data and effectively extract high-level features. Therefore, the traditional ML models (SVR and BPNN) are no longer suitable for complex time series data prediction tasks.

Finally, the prediction curve for a day is plotted by randomly selecting 0:00 on a day from the test set as the prediction moment point, as shown in Figure 10. The prediction results of these models are basically consistent with the trend of the true value curve, but the RGA model fits the best. Then, in order to show the comparison of the prediction performance for a longer period of time, we again randomly selected a certain day at 0:00 as the prediction moment point and plotted the prediction curves for a week, as shown in Figure 11. As can be seen in Figure 11, it is clear that the RGA model is closer to the true value, which is powerful proof of its robustness. It is worth noting that the RGA model has some errors in the prediction of some trough areas, which may be caused by the intensive ship traffic flow in summer and the wave wake impacting the buoy or the increase in rainfall, the rising water level and the increasing trend of reciprocal flow, but in general, the prediction results are basically consistent with the trend of the real values, and the fitting performance is good, which allows it to analyze the change pattern of drift more accurately.

In summary, the experiments in this section evaluate the prediction performance of nine network models in a short-term prediction scenario, which includes other methods of buoy drift prediction developed in recent years (as part of the benchmark model); however, the proposed RGA model presents the most satisfactory capability in inland multi-functional buoy drift prediction, and it also has a strong robustness to meet the demand of tracking its position in real time, which can reduce ship navigation accidents and improve navigation efficiency to a certain extent, and therefore, has more practical application value.

## 5. Conclusions and Future Works

In this paper, a deep neural network was applied to the drift prediction of a multi-functional buoy in an inland river, and a drift prediction model based on the combination of the attention mechanism and ResNet-GRU is proposed. In this model, the local features are extracted by the ResNet network, which has a significantly higher performance and stronger feature representation capability than CNN without a “skip connection”. The output features are used as the input of the GRU to capture time series waveform features, and then the attention mechanism is used to calculate the attention scores of hidden states. The importance of different features is dynamically determined, and greater weights are given to the important features. In addition, the proposed RGA model has the advantage of capturing both local and global dependencies and is able to adapt to multiple time steps of prediction. Experimental results show that the RGA model outperforms all other comparative models in predicting multi-functional buoy drift, with stronger robustness and generalization ability. Therefore, this study can be applied to the drift prediction of other similar types of buoys on the Yangtze River waters. However, this study also has some limitations:The attention mechanism used in this study is a spatial attention mechanism that calculates the importance of different features, and in fact, the target sequence also has different relevance in different time periods;In fact, the drift is also influenced by factors such as water level values, etc. In addition, due to the limited positioning accuracy, the RGA model may not be able to fully and accurately predict the drift; i.e., there is a certain amount of error.

In future work, we will add the time attention mechanism to calculate the importance of different time periods for drift while further exploring other factors affecting drift, looking for water level acquisition sensors with more stable performance, adding water level values and other influencing factors, improving the positioning accuracy of multi-functional buoys and continuing to study more effective neural network methods to enhance the stability and accuracy of the model in multi-functional buoy drift prediction so as to provide a decision basis for safe and stable ship navigation or reasonable navigation planning.

## Figures and Tables

**Figure 1 sensors-22-05120-f001:**
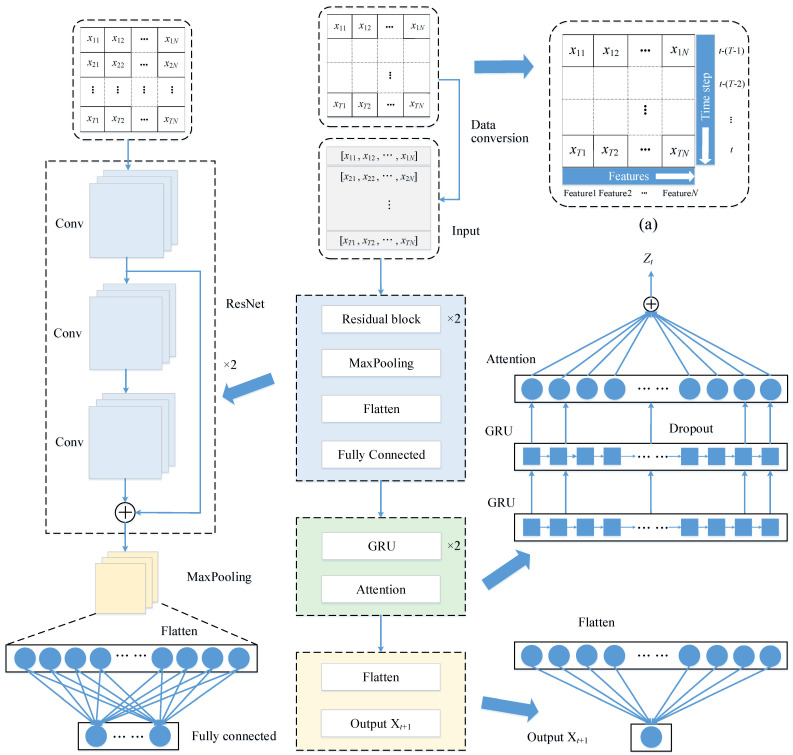
RGA model architecture. (**a**) The grid form of data.

**Figure 2 sensors-22-05120-f002:**
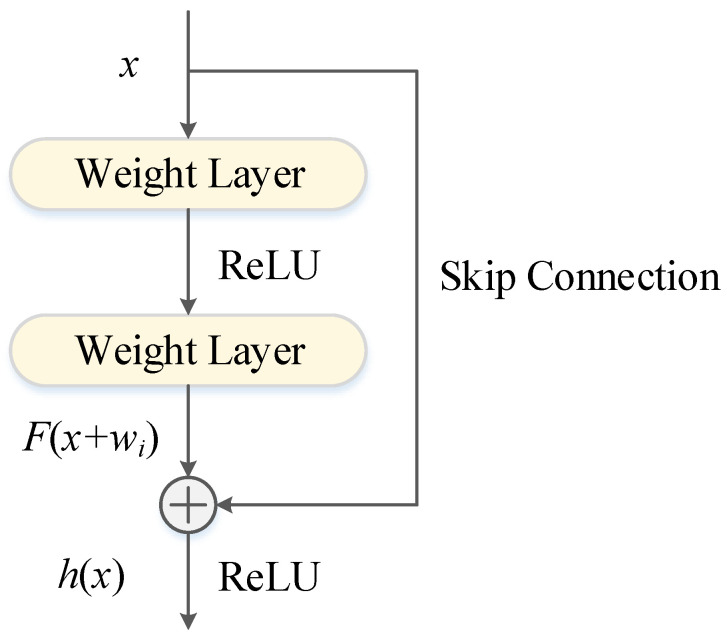
Residual block structure.

**Figure 3 sensors-22-05120-f003:**
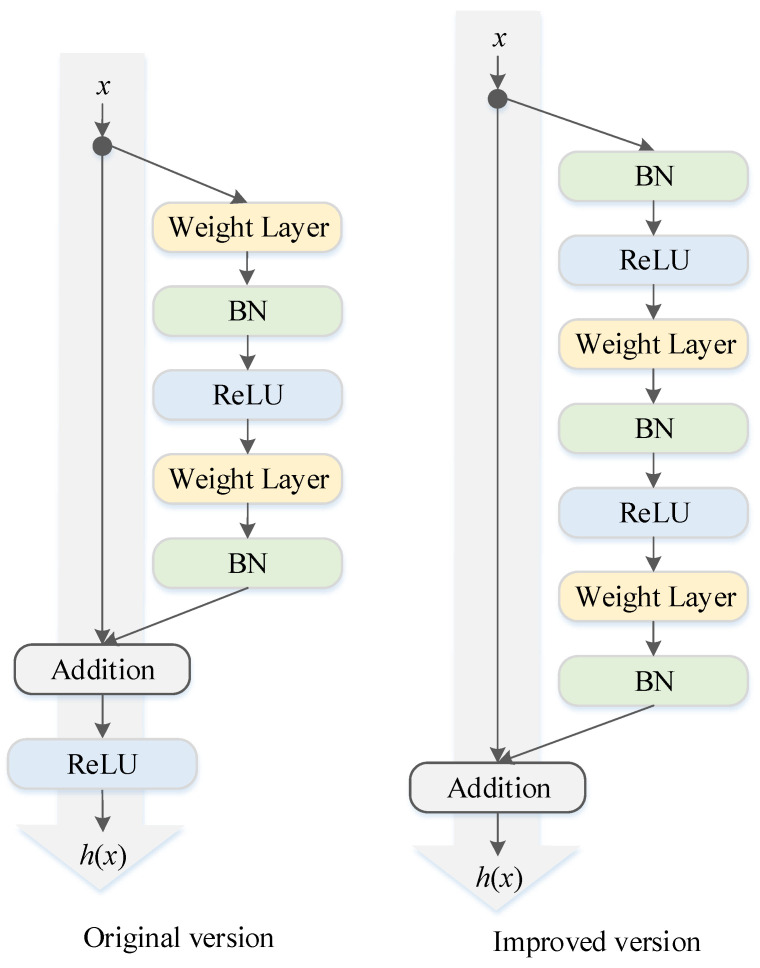
Comparison of original and improved residual blocks.

**Figure 4 sensors-22-05120-f004:**
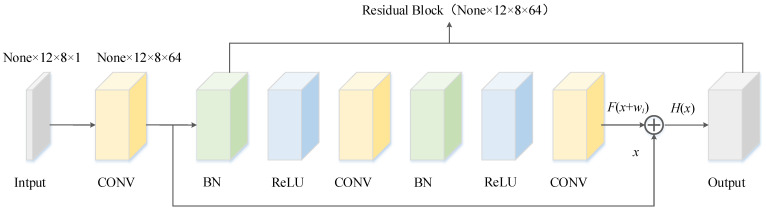
Dimensionality of each layer of the residual block.

**Figure 5 sensors-22-05120-f005:**
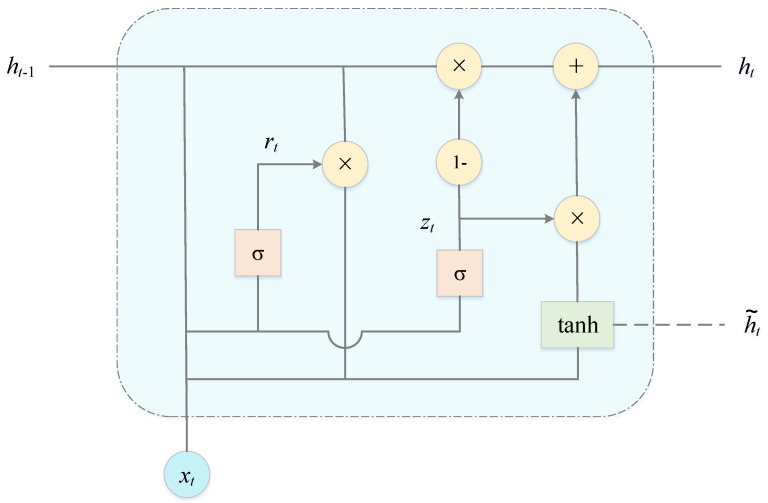
The structure of the GRU unit.

**Figure 6 sensors-22-05120-f006:**
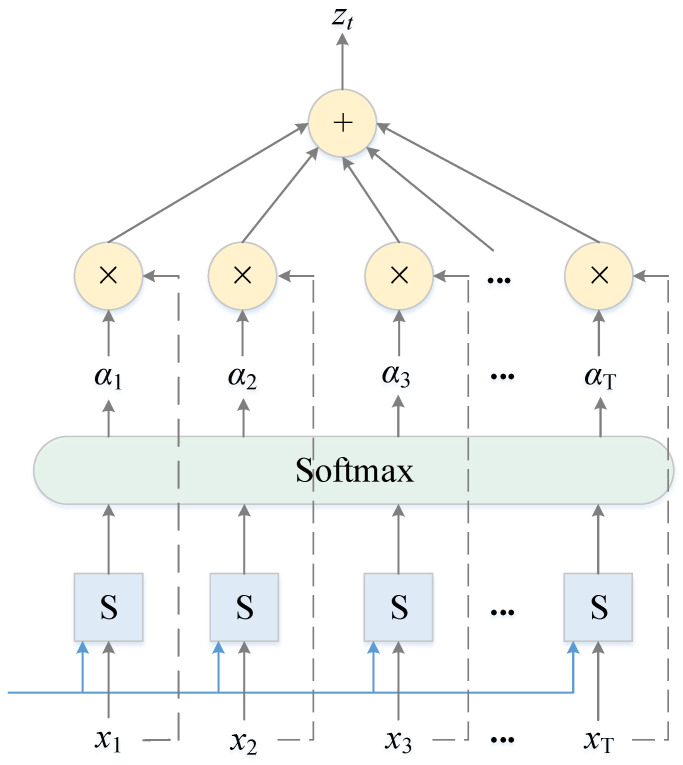
Structure of the attention layer.

**Figure 7 sensors-22-05120-f007:**
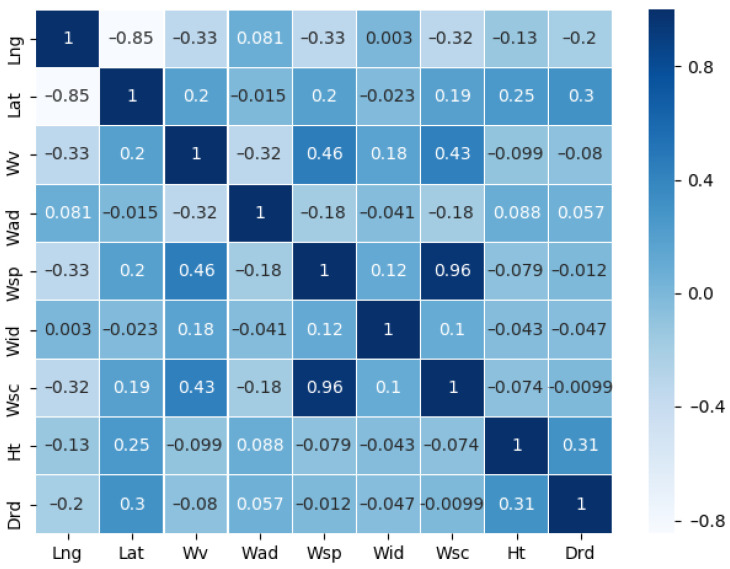
Results of Pearson correlation analysis between eigenvalues.

**Figure 8 sensors-22-05120-f008:**
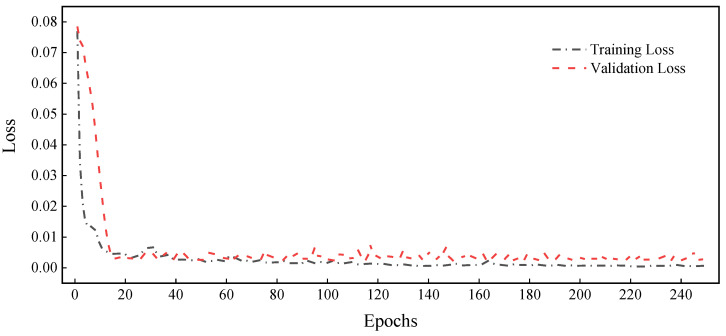
The curves of training loss and validation loss.

**Figure 9 sensors-22-05120-f009:**
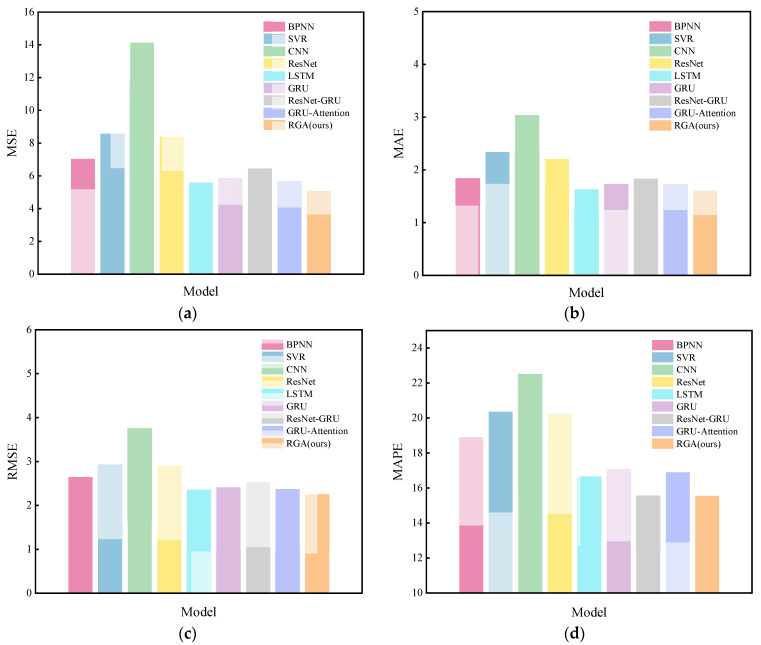
Comparison of prediction performance of different models: (**a**) MSE; (**b**) MAE; (**c**) RMSE; and (**d**) MAPE.

**Figure 10 sensors-22-05120-f010:**
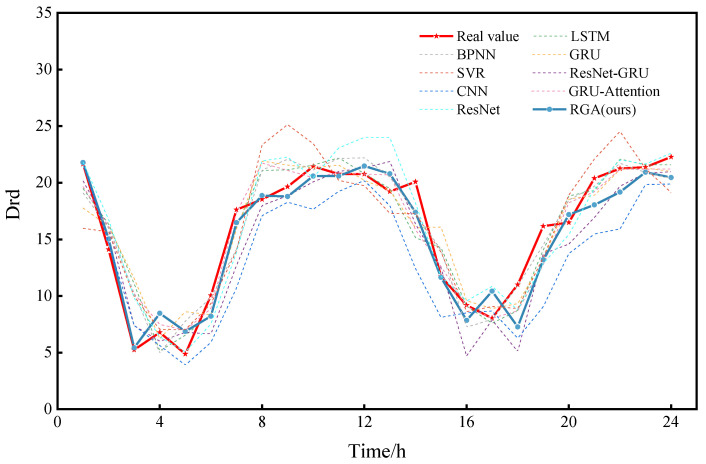
Comparison between RGA model and other models (one day).

**Figure 11 sensors-22-05120-f011:**
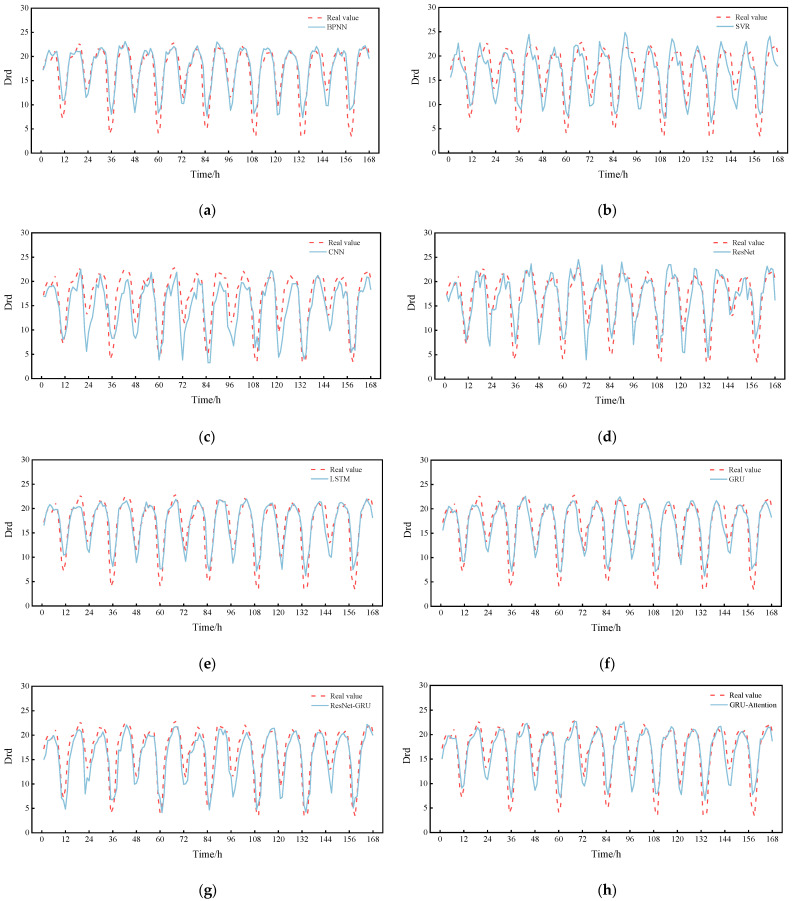
Comparison between RGA model and other models (a week). (**a**) BPNN; (**b**) SVR; (**c**) CNN; (**d**) ResNet; (**e**) LSTM; (**f**) GRU; (**g**) ResNet-GRU; (**h**) GRU-Attention; (**i**) RGA(ours).

**Table 1 sensors-22-05120-t001:** Abbreviation and unit of each variable.

Variable Name	Abbreviation	Unit	Variable Name	Abbreviation	Unit
Longitude	Lng	°	Wind direction	Wid	\
Latitude	Lat	°	Wind scale	Wsc	\
Water velocity	Wv	m/s	Tidal value	Ht	cm
Water direction	Wad	\	Drift value	Drd	m
Wind Speed	Wsp	m/s			

**Table 2 sensors-22-05120-t002:** The hardware and software platform configurations.

Configuration Name	Version Parameters
CPU	AMD Ryzen 5 3500X 6-Core Processor@3.60GHz
GPU	NVIDIA GeForce GTX 1050 Ti
RAM	16 G
Operating system	Ubuntu18.04.5
Programming languages	Python3.7.11
Deep learning framework	TensorFlow2.3.0
Other libraries	Keras2.4.3, scikit-learn0.24.2, etc.

**Table 3 sensors-22-05120-t003:** Parameter settings of the ResNet.

Model Parameters	Method/Values
Time step /h	12
Convolution layer feature maps of residual block 1/2	64/128
Convolution layer kernel size of residual block 1/2	3 × 3
Convolution layer strides of residual block 1/2	1
Max-pooling layer kernel size	2 × 2
Convolution layer activation function	Relu
Fully connected layer activation function	Relu
Loss function	MSE
Regularization method	BN
Optimization iteration algorithm	Adam

**Table 4 sensors-22-05120-t004:** Parameter settings of the GRU.

Model Parameters	Method/Values
Time step /h	12
Number of neurons in GRU layer 1/2	128/128
GRU layer activation function	Sigmoid and tanh
Dropout ratio in GRU layer 2	0.2
Loss function	MSE
Optimization iteration algorithm	Adam

**Table 5 sensors-22-05120-t005:** Comparison of prediction performance of different models.

Category	Model	Test			
		MSE	MAE	RMSE	MAPE
ML	BPNN	7.067512	1.846344	2.658479	18.916563
	SVR	8.602795	2.347254	2.933052	20.380289
DL	CNN	14.159271	3.046654	3.762880	22.531536
	ResNet	8.408451	2.213287	2.899733	20.260492
	LSTM	5.612912	1.640893	2.369158	16.677989
	GRU	5.889949	1.735634	2.426921	17.089307
	ResNet-GRU	6.480282	1.841023	2.545639	15.579802
	GRU-Attention	5.689599	1.732793	2.385288	16.924782
	**RGA (ours)**	**5.113036**	**1.609969**	**2.261202**	**15.575886**

## Data Availability

Owing to the nature of this research, the participants of this study did not agree for their data to be shared publicly; data are only available upon reasonable request.

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
