# Peer review of "Short-Term Drift Prediction of Multi-Functional Buoys in Inland Rivers Based on Deep Learning"

_sensors, 2022, doi:10.3390/s22145120_

Round 1

Reviewer 1 Report

The authors of this paper propose a hybrid prediction model for short-term drift prediction of inland buoys based on deep learning. They also compared performance results of several models. This is interesting work and the paper is well written as well as easy to follow. 

However, I will suggest authors provide an more extensive discussion about the application of the work.

Author Response

Dear Reviewer:

We appreciate very much the referee’s time and effort to review our paper and the in-depth and valuable comments. We have considered these comments and incorporated them fully into the revised manuscript, which, we believe, has improved the quality of the paper.

Point 1: I will suggest authors provide an more extensive discussion about the application of the work.

Response 1: Agreed and is a very valuable suggestion for us to improve the quality of the paper. We add a discussion of the application of this work in the experimental results and discussion section in Section 4.5. “In summary, the experiments in this section evaluate the prediction performance of nine network models from a short-term prediction scenario, which contains other methods of buoy drift prediction in recent years (as part of the benchmark model). However, the proposed RGA model presents the most satisfactory capability in inland multi-functional buoy drift prediction, and it also has a strong robustness to meet the demand of tracking its position in real time, which can reduce ship navigation accidents and improve navigation efficiency to a certain extent, and therefore, has more practical application value.”. In addition, this study can be applied to the drift prediction of other similar types of buoys on the Yangtze River waters.

Action 1: In the revised manuscript, we add a more detailed description about the application of the work in Sections 4.5 and 5 according to the reviewer’s comments.

Besides, in the revised manuscript, we tried our best to improve the manuscript and made some changes in the manuscript. These changes will not influence the content and framework of the paper. And here we did not list the changes but highlighted in red in revised paper. Finally, we still want to thank the anonymous referees for your hard work which has greatly improved the original manuscript, and hope that the correction will meet with approval.

Once again, thank you very much for your comments and suggestions.

Your sincerely

Fei Zeng, Hongri Ou and Qing Wu

Reviewer 2 Report

Various parts of the manuscript could be improved:

- Use of the English language in the manuscript should be revised. Personal pronouns, such as "we" or "our", should not be used in the manuscript. In the manuscript neutral and impersonal language should be used.

- I would recommend to add a couple more keywords. Maybe something like Waterway or Ship navigation.

- In the end of the introduction the aim of this work should be named very clearly. 

- What is actual authors contribution to the model, described in section 2? In my opinion, the model development process could be explained more in detail.

- Regarding the data acquisition, it is unclear why multi-functional buoy #4 at Nantong anchorage No. 2 near the main waterway of the Yangtze River is selected as the object? This selection somehow should be properly supported. In general, entire data acquisition process should be more clearly defined and supported.

- Major drawback of the manuscript – analysis part. As it is now, performed analysis is too limited. It should be performed in more detailed way.

Author Response

Dear Reviewer:

We appreciate very much the referee’s time and effort to review our paper and the in-depth and valuable comments. We have considered these comments and incorporated them fully into the revised manuscript, which, we believe, has improved the quality of the paper.

Point 1: Use of the English language in the manuscript should be revised. Personal pronouns, such as "we" or "our", should not be used in the manuscript. In the manuscript neutral and impersonal language should be used.

Response 1: Agreed and is a very valuable suggestion for us to improve the quality of the paper. We have re-examined the original manuscript and deleted all the personal pronouns in active sentences.

Action 1: In the revised manuscript, we have checked the English language and style carefully and modified all the sentences to ensure no personal pronouns are used in this sentences.

Point 2:  I would recommend to add a couple more keywords. Maybe something like Waterway or Ship navigation.

Response 2: This is indeed a very detailed review and valuable suggestion.

Action 2: In the revised manuscript, we have added "inland waterway" and "ship navigation" in the keywords field.

Point 3: In the end of the introduction the aim of this work should be named very clearly. 

Response 3: This is indeed a very detailed review and valuable suggestion. The main purpose of this work as follows.1) To construct the structure of the RGA hybrid model and give optimal parameter settings, and then to improve the prediction accuracy of short-term drift of inland waterway buoys; 2) To investigate the effects of different hydro-meteorological parameters on the variation of buoy position drift and select appropriate input features for the prediction model; 3) To evaluate the performance of RGA model put forward in this paper by comparison of the results with other baseline models.

Action 3: In the revised manuscript, we have added  clear explanation about the aim of this work in the end of the introduction according to the reviewer’s comments.

Point 4: What is actual authors contribution to the model, described in section 2? In my opinion, the model development process could be explained more in detail.

Response 4: Agreed and is a valuable suggestion. At present, most of available drift prediction models of multi-function buoys are established by using of traditional shallow neural networks, such as BPNN and RBF, or used a single recurrent neural network (RNN) variant series. Nevertheless, it is not well adapted to the situation because the prediction models based on traditional shallow neural networks could not capture the correlation in time series data and based on RNN could not extract relevant high-level abstract features effectively. Presently, the fusion of multiple models can significantly improve the prediction accuracy in many areas, except for buoy drift prediction. Considering the advantages of excellent local feature extraction of convolutional neural networks (CNN), and the strong times series memory capability of GRU (RNN series network), it is feasible to establish the hybrid model approach based on CNN and GRU  for buoy drift prediction. Therefore, we stack the improved residual block structure to form a ResNet layer, which is one of a CNN series network. Then a convolutional layer is added after the input layer to make the input number is consistent with the output number of the residual block in the forward propagation process. In addition, an attention mechanism is also added for adaptive weight assignment to the features. It can highlight the attention to important information rather than assigning weights randomly when we make the prediction of the buoy drift at a future moment. Taking the above considerations into account, a model that incorporates ResNet, GRU and attention mechanism is proposed. It can improve the prediction accuracy of buoy drift significantly due to the advantages of ResNet, GRU and attention mechanism.

Action 4: We have added a clear explanation about the model development process in section 2.

Point 5: Regarding the data acquisition, it is unclear why multi-functional buoy #4 at Nantong anchorage No. 2 near the main waterway of the Yangtze River is selected as the object? This selection somehow should be properly supported. In general, entire data acquisition process should be more clearly defined and supported.

Response 5: This is indeed a very detailed review and valuable suggestion. There are two reasons why we choose multi-functional buoy #4 at Nantong anchorage No. 2 near the main waterway of the Yangtze River as the object.

Firstly, Yangtze River is one of the busiest waters in the world. As shown in the figure below, the buoy we arranged is located in Nantong section of Yangtze River, which is near the mouth of Donghai Sea in Shanghai. And it is also the area of the confluence of Tonglu Canal and Yangtze River. So it has large ship traffic volume and complex navigation environment. Especially at night or in bad weather, a variety of complicated ships will reach  anchorage No. 2 in Nantong for anchoring, which increases the risk of mutual collision. Therefore, it is practical and reasonable for us to arrange multi-functional buoy at Nantong anchorage No. 2 to ensure the safety of ship navigation. Due to the multi-functional buoy #4 is closer to the tidal observation station of Tiansheng Harbor, we could collect the real tidal values close to the multi-functional buoy #4 for one of input variables of out model.

Secondly, we signed a project with Shanghai Maritime Bureau. This project need us to place the multi-functional buoy #4 at Nantong anchorage No. 2 near the main waterway of the Yangtze River. At the same time, we also added experimental instruments on the multi-functional buoy #4 to get hydro-meteorological data for our study subjects.

Action5: We have added a more clear detail for the data collection process according to the reviewer’s suggestion in section 3.1 of the revised manuscript.

Point 6: Major drawback of the manuscript-analysis part. As it is now, performed analysis is too limited. It should be performed in more detailed way.

Response6: Agreed and is a very valuable suggestion for us to improve the quality of the paper. In the model parameter settings of section 4, the initial parameter settings of the ResNet layer of our proposed RGA model mainly refer to the parameters of the ResNet network model developed by Kaiming He. Owing to our data set is much smaller than other application scenarios, so the number of residual block stacking was set to 2. In addition, we compare the parameter settings of multiple ResNet layers and GRU layers, and also add the regularization method and Dropout for preventing model overfitting, find the optimal parameters through multiple training, and then save the weights of the model for prediction, All parameters given in this paper are optimal cases. In the Experimental Results and Discussion section, we have modified the analysis of experimental results section in more detail and added a discussion of this work in terms of applications. The reviewer’s suggestion motivates us to do more in-depth research.

Action6: In the revised manuscript, we add a more detailed description about the settings of model parameters in sections 4.2 and 4.3. We have added a clear explanation about the experimental results and the discussion of this work in section 4.5.

Besides, in the revised manuscript, we tried our best to improve the manuscript and made some changes in the manuscript. These changes will not influence the content and framework of the paper. And here we did not list the changes but highlighted in red in revised paper. Finally, we still want to thank the anonymous referees for your hard work which has greatly improved the original manuscript, and hope that the correction will meet with approval.

Once again, thank you very much for your comments and suggestions.

Your sincerely

Fei Zeng, Hongri Ou and Qing Wu

Round 2

Reviewer 2 Report

Thank You for Your detailed responses. However, I still have one more comment regarding the the data acquisition. 

In Your answers You explained why multi-functional buoy #4 at Nantong anchorage No. 2 near the main waterway of the Yangtze River were selected as the object. However, in the manuscript this selection is still not explained. Respectively, this selection also should be explained in the manuscript for the readers.

Author Response

Dear Reviewer:

We appreciate very much the referee’s time and effort to review our paper and the in-depth and valuable comments. We have considered these comments and incorporated them fully into the revised manuscript, which, we believe, has improved the quality of the paper.

Point 1: In Your answers You explained why multi-functional buoy #4 at Nantong anchorage No. 2 near the main waterway of the Yangtze River were selected as the object. However, in the manuscript this selection is still not explained. Respectively, this selection also should be explained in the manuscript for the readers.

Response 1: Agreed and is a very valuable suggestion for us to improve the quality of the paper.

Action 1:  In the revised manuscript, we have added the explains why multi-functional buoy #4 at Nantong anchorage No. 2 near the main waterway of the Yangtze River were selected as the object in Section 3.1.

Besides, in the revised manuscript, we tried our best to improve the manuscript and made some changes in the manuscript. These changes will not influence the content and framework of the paper. And here we did not list the changes but highlighted in yellow in revised paper. Finally, we still want to thank the anonymous referees for your hard work which has greatly improved the original manuscript, and hope that the correction will meet with approval.

Once again, thank you very much for your comments and suggestions.

Your sincerely

Fei Zeng, Hongri Ou and Qing Wu
